# Impact of Galectin-3 Circulating Levels on Frailty in Elderly Patients with Systolic Heart Failure

**DOI:** 10.3390/jcm9072229

**Published:** 2020-07-14

**Authors:** Klara Komici, Isabella Gnemmi, Leonardo Bencivenga, Dino Franco Vitale, Giuseppe Rengo, Antonino Di Stefano, Ermanno Eleuteri

**Affiliations:** 1Department of Medicine and Health Sciences, University of Molise, Via Francesco de Sanctis, 1, 8610 Campobasso, Italy; 2Pulmonary Rehabilitation Unit and Laboratory of Cytoimmunopathology of the Heart and Lung, Istituti Clinici Scientifici Maugeri, 28010 Veruno, Italy; isabella.gnemmi@icsmaugeri.it (I.G.); antonino.distefano@icsmaugeri.it (A.D.S.); 3Department of Translational Medical Sciences, University of Naples Federico II, 80131 Naples, Italy; leonardobencivenga@gmail.com (L.B.); giuseppe.rengo@unina.it (G.R.); 4Department of Advanced Biomedical Sciences, University of Naples “Federico II”, 80131 Naples, Italy; 5Clinica San Michele, Via Appia 187, Maddaloni, 81024 CE, Italy; dinofranco.vitale@fastwebnet.it; 6Istituti Clinici Scientifici Maugeri SpA Società Benefit (ICS Maugeri SpA SB), Telese Terme, 82037 BN, Italy; 7Division of Cardiology, Istituti Clinici Scientifici Maugeri, 28010 Veruno, Italy; ermanno.eleuteri@icsmaugeri.it

**Keywords:** frailty, heart failure, galactin-3, elderly, biomarker

## Abstract

Background: Heart Failure (HF), a leading cause of morbidity and mortality, represents a relevant trigger for the development of frailty in the elderly. Inflammation has been reported to play an important role in HF and frailty pathophysiology. Galectin-3 (Gal-3), whose levels increase with aging, exerts a relevant activity in the processes of cardiac inflammation and fibrosis. The aim of the present study was to investigate the potential of Galectin-3 to serve as a biomarker of frailty in HF patients. Methods: 128 consecutive patients aged 65 and older with the diagnosis of systolic HF underwent a frailty assessment and blood sample collection for serum Gal-3 detection. A multivariable regression analysis and decision curve analysis (DCA) were used to identify significant predictors of frailty. Results: Frailty was present in 42.2% of patients. Age: Odds Ratio (OR) = 3.29; 95% Confidence Interval CI (CI) = 1.03–10.55, Cumulative Illness Rating Scale Comorbidity Index (CIRS-CI): OR = 1.85; 95% CI = 1.03–3.32, C-Reactive phase Protein (CRP) OR = 3.73; 95% CI = 1.24–11.22, N-terminal-pro-Brain Natriuretic Peptide (NT-proBNP): OR = 2.39; 95% CI = 1.21–4.72 and Gal-3: OR = 5.64; 95% CI = 1.97–16.22 resulted in being significantly and independently associated with frailty. The DCA demonstrated that the addition of Gal-3 in the prognostic model resulted in an improved clinical ‘net’ benefit. Conclusions: Circulating levels of Gal-3 are independently associated with frailty in elderly patients with systolic HF.

## 1. Introduction

Frailty is a geriatric syndrome characterized by a multidimensional and cumulative decline in many organs and systems, thus contributing to increased vulnerability to stressors and negative outcomes [1]. Frailty is considered an ultimate consequence of accelerated aging, with a wide and heterogenous range of multi-functional declines defining it as a distinct biological syndrome [2]. Epidemiological data reveal that about 7 million Americans are frail [3], while in the European countries the estimated prevalence of frailty in overall populations is about 18%, ranging from 12% in community-based studies to 45% in non-community settings [4]. The pathophysiological mechanisms of frailty are not established yet; however, oxidative stress, enhanced inflammation, immune system dysfunction, endocrine dysregulation, neuro-hormonal axis alternations and metabolic imbalance are considered possible pathways of frailty onset and development [5]. Heart failure (HF), extremely prevalent in the elderly population, is associated with high medical costs and is a leading cause of hospitalizations and adverse events [6]. Within the elderly patient population, HF represents a relevant trigger for vulnerability and frailty development. Furthermore, frailty is particularly common in HF patients, with a prevalence ranging from 30 to 52% [7,8]. The burden of frailty in hospitalized patients with HF is described as accounting for 56 to 76%, higher than the frailty prevalence in community or non-community elderly without HF [9]. The wide range of prevalence mainly depends on the tools adopted for the assessment and on HF classes. These are defined according to echocardiographic left ventricle ejection fraction (LVEF) values as HF with reduced, mid-range or preserved LVEF. The Clinical Frailty Scale (CFS) is a validated tool for the assessment of frailty, predominantly based on clinical judgment, that has been correlated with hospitalizations and survival in the elderly population [10]. Furthermore, CFS has shown good effectiveness in identifying frailty in HF patients [7]. Among the several variables associated with clinical frailty, including age, comorbidities, nutritional status and cognitive function, HF severity also exerts a relevant role. In advanced HF, the prevalence of frailty is higher [11], contributing to an even more challenging management and a worse outcome. Moreover, an important pathophysiological overlap between HF and frailty has been described [7,8]. Inflammation has been reported to be a key regulator of HF onset and progression, and the identification of cardio-inflammatory phenotypes among HF patients may have relevant future implications, even regarding the therapeutic approaches [12]. Hypoperfusion and chronic congestion can contribute to tissue hypoxia, cellular apoptosis or necrosis, and the upregulation of inflammatory pathways. In turn, inflammation has also been associated with the pathophysiology of frailty, mainly through the activation of metabolic pathways and a direct influence on protein degradation [13]. Biomarkers are routinely used in clinical practice for the diagnosis, safety and efficacy monitoring of medical interventions or risk stratification. It has been assumed that the molecular mechanisms involved in the so-called phenotype of physiological aging and frailty are generally comparable, but they differ regarding the degree of modulation [14]. Current literature emphasizes the need for the identification of biomarkers involved in frailty development, and several studies have reported that C-reactive phase protein (CRP), IL-6 and brain natriuretic peptide (BNP) are associated to pre-frailty and frailty, even in HF patients [7,15,16]. Galectin-3 (Gal-3), a β-galactoside binding lectin, plays a significant role in systemic inflammation, cardiac fibrosis and HF progression [17,18]. Elevated Gal-3 levels are associated with negative outcomes, while the inhibition of Gal-3 has been described as improving cardiac remodeling in experimental models of systolic HF [19]. Gal-3 levels increase with age and have been associated with comorbidities [20]. Of note, the permanent replacement of functional tissue by fibrosis is a characteristic of the aging process, and furthermore it is considered, together with inflammation, one of the hallmarks of aging. A recent systematic review, focused on the identification of novel biomarkers related to frailty, suggested the provision of a more extensive exploration of inflammatory and fibrosis biomarkers in frailty cohorts, besides experimental models [21]. Therefore, considering the role of Gal-3 in the systemic inflammation and cardiac fibrosis process, we sought to investigate the association between Gal-3 and frailty in a population of elderly HF patients.

## 2. Materials and Methods

### 2.1. Study Population

We enrolled 128 consecutive elderly HF patients with systolic dysfunction, admitted at the Cardiac Rehabilitation Unit of the Salvatore Maugeri Foundation IRCCS, Institute of Veruno, Italy. The inclusion criteria were: (a) age ≥ 65 years; (b) diagnosis of HF from at least six months, due to ischemic or non-ischemic etiology; (c) LVEF ≤40%; (d) stable clinical conditions for at least one month before enrollment; and (e) guidelines-based optimal medical therapy. The exclusion criteria listed: (a) chronic inflammatory diseases or ongoing infectious diseases; (b) severe renal or hepatic function impairment; (c) malignancies; (d) psychiatric disorders; (e) terminally ill patients; and (f) patients not able to understand the scope and methods of the study and/or to sign the informed consent. Patients after cardiac surgery or without systolic disfunction were not considered for this study in order to avoid heterogeneity in the population characteristics. At the time of enrollment, all patients underwent a complete clinical examination (including an assessment of the New York Heart Association (NYHA) functional class and an echocardiography to evaluate LVEF, a structured interview to collect data regarding demographic features, cardiovascular risk factors, comorbidities and HF medications, and a blood draw to laboratory tests, including a full blood count, creatinine, electrolytes, CRP, Gal-3 and N-terminal -pro-Brain Natriuretic Peptide (NT-proBNP). CRP and NT-proBNP were measured according to standardized laboratory methods and were considered as possible comparable variables with respect to Gal-3, constituting parameters widely used in clinical practice and research. The Glomerular Filtration Rate (GFR) was calculated according to the GFR Chronic Kidney Disease Epidemiology Collaboration equation [22]. The study was carried out in conformity with the 1975 Declaration of Helsinki, and it has been approved by the Local Ethical Committee, code 2046 CE. Informed consent was obtained from each subject prior to entering the study.

### 2.2. Frailty Assessment

CFS was adopted as a clinical tool for the frailty assessment. This scale includes the evaluation of several domains, such as cognition, mobility, function and co-morbidities, through a direct examination of history and medical records [10,23]. According to their functional capacity, level of dependence and comorbidities, patients are scored from 1 to 9 points: scoring 1–4 classifies non-frail individuals, whereas 5–8 classifies frail patients, and a score of 9 identifies terminally ill patients. In our population, terminally ill individuals approaching the end of life were not included. The Cumulative Illness Rating Scale (CIRS) was also administered to all patients, and the CIRS-Comorbidity Index (CIRS-CI) was calculated based on the count of the organ system with moderate to greater impairment [24,25].

### 2.3. Biomarker Assessments

To detect the Gal-3 circulating levels, an additional blood sample was collected after 12 h of fasting using Becton Dickinson Vacutainer Cat Plus REF 367,896 for serum (BD Diagnostics, Franklin Lakes, NJ, USA). The serum was then aliquoted and immediately frozen at −80 °C until analysis. An enzyme-linked immuno-absorbent assay (ELISA) quantification (Modular Analytics, Roche Diagnostics, Mannheim, Germany) was used for the Gal-3 measurement, and the manufacturers’ instructions were carefully followed for each of the ELISA kits used [26,27]. 

### 2.4. Statistical Analysis

Continuous variables are expressed as the mean ± standard deviation, while categorical variables are expressed as the number of events and percentages. According to the variable distribution normality, Student’s t-tests were used to compare the means of the groups defined by the absence or the presence of frailty. For categorical variables, a chi-square test was performed. A multivariable logistic regression analysis was used to identify the factors associated with frailty. Taking into account the study sample size, the rule of “at least five observed events for each tested variable” [28] and parsimonious criteria, the following 10 factors were selected as possible confounders to test the possible association of Gal-3 levels with the probability of frailty presence in the logistic model; age, gender, body mass index (BMI), Chronic Kidney Disease (CKD) (GFR ≤ 50 mL/min), NYHA class III or greater, LVEF, CIRS-CI, hemoglobin, CRP and logarithmic transformation of NT-proBNP. Given the skewed data distribution, a logarithmic transformation was applied for NT-proBNP before all computations in order to approximately conform to a normal distribution [29]. To have a biological meaning, the odds ratios (OR) of the significant continuous factors such as CRP, CIRS-CI, Gal-3 and ln-NT-proBNP are expressed relative to a 1 SD variation, while the OR of the age is expressed relative to a 10 years variation. The statistical significance (by stepwise selection) and the functional form (linearity and non-linearity) of all factors were checked and modeled using a multivariable fractional polynomial (MFP) algorithm [30]. The relative weight of each significant factor was estimated by measuring the partial contribution of each variable to the global explained variation of the frailty presence in the studied population, as measured by the pseudo R2 (pR2), using the Shapley–Owen decomposition algorithm. A linear regression analysis was performed to assess the correlation between Gal-3 levels and CSF score values. A Decision Curve Analysis (DCA) was employed to evaluate the potential utility of using the final frailty prognostic model as a decision maker tool (‘clinical action’) in the clinical environment [30,31,32]. Notably, this task was accomplished without the commitment of performing the ad hoc studies required to obtain the information relevant to the several patient management clinical scenarios potentially conditioned by a frailty status assessment. DCA computes the ‘net’ benefit obtained by applying the investigated prognostic model to make a clinical decision. In this context, ‘make a clinical decision’ refers to any (not-specified) clinical decision or action taken on patients (‘treatment’) as a function of the results of the prognostic frailty model. In order to assess the additive clinical utility gained by the inclusion of the Gal-3 levels in the final logistic model, the ‘net’ benefit curves obtained by DCA on two prognostic models (one with and one without Gal-3) were compared. Both curves are compared to the ‘net’ benefit attained by the two other possible clinical actions; namely, applying the given ‘clinical decision’ to all patients (‘treat all’) or to any patients (‘treat none’). The clinical benefit is termed ‘net’ since it is assessed using a metric that takes into account the “cost” as well as the “benefits” of acting according to a given prognostic model. All analyses were performed with a 0.05 type I error threshold using STATA 14.1 software (Stata Corp. LP College Station, TX, USA).

## 3. Results

### 3.1. Characteristics of the Overall Study Population and of Subgroups According to Frailty Status

The characteristics of the overall study population, stratified by the presence or the absence of frailty, are reported in Table 1. The study population, including 128 elderly HF patients, presented a mean age and a mean LVEF of 69.2 ± 4.8 years and 28.7 ± 8.5%, respectively. The male gender was more prevalent, with 87.5% (112), and the mean BMI was 25.4 ± 4.3. The studied population was stratified into three groups according to the CSF score (group 1 no frail, CSF ≤ 4, 54 (42.2%) patients; group 2 mildly frail, CFS = 5, 39 (30.5%) patients; group 3 moderately/severe frail, CFS ≥ 6, 35 (27.3%) patients). Groups 2 and 3 were considered together for the logistic analysis, and their joint characteristics are reported in Table 1. In the univariate analysis, the two groups (frail vs. non-frail) were homogenous for gender, cardiovascular risk factors, HF therapies and laboratory tests, including white blood cells, fibrinogen and sodium level. Frail patients presented a trend, although not a significant one, of BMI reduction (24.5 ± 4.6 vs. 26.0 ± 4.2 kg/m^2^; *p* = 0.07). Compared to non-frail patients, frail patients were older (*p* = 0.008), with a worse kidney function (*p* = 0.001), and they had lower hemoglobin levels (*p* = 0.001). Importantly, frail patients showed worse LVEF (*p* = 0.02), a worse NYHA functional class (*p* ≤ 0.0001), and higher NT-proBNP levels (*p* = 0.002). Of note, higher CRP (*p* < 0.0001) and Gal-3 levels (*p* < 0.0001) characterized frail patients. The linear correlation analysis of Gal-3 vs. CFS revealed a significant (*p* < = 0.0001) progressive increment of the biomarker levels associated with a CFS score increase, with an estimated 5.7 ng/mL increase of Gal-3 for one unit increase of the score. A uniform increment is assumed given the lack of significance of the non-linear relationship.

### 3.2. Clinical and Laboratory Factors Associated with Frailty

The multivariable logistic regression analysis (Table 2) indicated that age, CIRS-CI, CRP, NT-proBNP and Gal-3 were significantly and independently associated with frailty and that the functional form of their relationship was linear. The global strength of these associations was 50%, as defined by the global pseudo R2, thus indicating a robust global association. As documented by the percent contribution to the global pseudo R2, Gal-3 together with NT-proBNP, CRP and CIRS were the factors that correlated most with frailty. Of note, Gal-3 explained a relevant portion of the global association (R2 = 39.4%). From a clinical perspective, Gal-3 improved the net benefit of the predictive model. Indeed, as documented by the decision curve analysis (Figure 1), the full model showed a clinical net benefit profile that was higher than the model that did not include Gal-3 along the entire relevant span (20–60%) of the decision threshold probability. Of note, both models exhibited profiles above those observed with the “treat all” and “treat none” strategies.

## 4. Discussion

The main finding of the present study is the identification of Gal-3 as a circulating marker that is significantly and independently associated with frailty in HF patients. Furthermore, we have demonstrated, through the application of DCA, the clinical relevance of Gal-3 in frail HF patients. In our study population of elderly HF patients, the prevalence of frailty, assessed through CFS, was 42%. Systematic reviews and meta-analyses have reported that almost half of patients with HF are affected by frailty, with a prevalence ranging from 18 to 54% depending on the intrinsic features of the population studied and the tools adopted for the frailty assessment [33,34]. In clinical and research practice, the most common methods used to assess frailty are: the Fried phenotype and Frailty Index. The Fried model is based on the evaluation of physiological reserves’ decline, such as: weight loss, weakness, poor endurance, slowness and a low physical activity level [35]. The Frailty Index is based on a multidisciplinary list of items encompassing information on symptoms, signs, laboratory results, comorbidity burden and activities of daily living. Although physical performance is only one feature of frailty and does not conclusively identify frailty, different scales such as gate speed, short physical performance battery and handgrip strength may help in the identification of individuals who warrant more specific frailty assessments. Different studies have confirmed both the predictive value of these instruments for a negative outcome of HF patients and also the association with a worse quality of life and functional impairment [36,37]. Otherwise, there are some challenges regarding the utility of these models in clinical practice: (a) the clinical manifestation of HF, in particular the physical performance, may overlap with most of the physical performance scales, including the Fried phenotype; (b) the health deficits evaluated by the Frailty Index, or Multidimensional Evaluations, may be related to acute HF, whereas the estimation of frailty may not be accurate; (c) the accumulation of health deficits may be related to the aging process and not necessarily to the frailty phenotype [37]. An ideal diagnostic tool should be characterized by a high sensitivity and high specificity, which means detecting few false negative and few false positive results. In community HF patients [7], the Fried phenotype model presented the highest sensitivity at 93% and the highest rate of false positive results at 14%. EFS reached the highest specificity at 98% and the highest false negative rate at 18%. The frailty index sensitivity in this population was 75%, while the rate of false negative results was 12%. The Fried phenotype and physical performance scales were characterized by false negative rates of around 3%. In our study, we employed CFS for the identification of frailty, which has been demonstrated as having a good sensitivity and specificity of 87% and 89%, while the false negative rates were 6% and the overall misclassification was 12% [7]. Moreover, our data on frailty prevalence are in line with those recently reported by Sze and colleagues [7], indicating that the prevalence of frailty evaluated with different assessment tools ranges from 27–47% in HF patients. In the present study, frail subjects were older, with a worse renal function, lower hemoglobin level and higher number of comorbidities. Importantly, in our multivariate model, we also introduced CIRS-CI, a risk adjustment tool useful for assessing patients’ comorbidities and prognoses [38,39], not included in the CFS, which came out as significantly associated with frailty. Regarding the parameters related to cardiac function, frail HF patients presented a worse NYHA functional class, worse LVEF and higher NT-proBNP levels. Furthermore, in the multivariate logistic model, NT-proBNP emerged as being independently associated to frailty. In addition, NT-proBNP has been suggested as a useful biomarker in the identification of frailty in multiple myeloma patients [40]. The relationship between inflammation, frailty and HF is extremely complex since both inflammation and HF incidences increase with age, and frailty is the most prevalent geriatric syndrome. However, CRP and IL-6 have been suggested as promising biomarkers of frailty in several cross-sectional studies [41,42]. Another study, focused on physical performance and frailty in HF patients, revealed that a high sensitivity CRP is associated with a lower physical performance and frailty in the HF population [43]. Thus, our results on inflammation, evaluated through CRP, are substantially in agreement with the above-mentioned studies. In this regard, it is important to underline that ongoing inflammatory or infection diseases represented exclusion criteria in our study; thus, the higher CRP values observed in frail subjects cannot be ascribed to the presence of these conditions. It should be mentioned that the specificity and sensitivity of the frailty identification of the above-mentioned biomarkers is not fully defined. Recently, Marzetti and colleagues [44] reported that 30 selected core inflammatory biomarkers provided a proportion of correct classification of physical frailty and sarcopenia of 75.6%, which in older women reached 76.5%, thus indicating a gender-specific signature. The incorrect classification of frailty among HF patients may produce dramatic consequences regarding the identification of patients who are at a higher risk of disability, adverse drug effects and clinical outcome. On the other hand, a more accurate frailty detection may facilitate targeted interventions in order to improve outcomes and reduce the frailty burden. Indeed, combinations of frailty assessment tools and biomarkers of cellular ageing, inflammation and immunosenescence have demonstrated an amelioration of the discriminatory power regarding adverse outcomes in frail populations [45]. In both univariate and multivariate analyses, we have shown that Gal-3 is significantly associated with frailty in the clinical setting of chronic HF. In medical literature and studies, multivariable regression models are widely used for the purpose of prediction or diagnosis. The diagnostic performance of these models is generally assessed by measuring the sensitivity/specificity or via a Receiver Operating Characteristic (ROC) curve analysis. These approaches, when transferred into clinical practice, use metrics that do not consider the balance between ‘harm and benefits’ or, when they do, use the same weight for both (1 harm equal to 1 benefit), or the metric itself has a meaning that is hard to transfer into clinical settings. For example, the area under the ROC curve is the probability of correctly diagnosing a *pair* of subjects founding on their scores (however, do clinicians diagnose patients as a couple?) [46]. In the attempt to give an idea of the potential clinical impact of the significant association that we found between Gal-3 and the frailty score, we performed a DCA analysis devised to overcome the mentioned limitations. The clinical net benefit achieved by adding the Gal-3 to the model as a prognostic marker (Figure 2) is a balance of ‘harm and benefits’ and shows higher values when compared to the deprived model along the full range of possible decision thresholds, i.e., the thresholds at which clinical decisions are taken and that may vary from patient to patient for a variety of reasons. Observing that the net benefit of the Gal-3 profile is also higher than the ‘treat all’ and ‘treat none’ outline led us to conclude that if the biomarker will be employed as a prognostic tool to drive the clinical management of patients, it has the potential to be a clinical benefit not only in comparison to the alternative strategies that involve managing all or none of the patients irrespective of Gal-3 values, but also in comparison to using a deprived prognostic model for patient management.

Gal-3 is a multifunctional protein implicated in different physiological processes, and it is mainly considered as a macrophage activation marker. Based mainly on experimental models, Gal-3 appears to play a key role in embryo implantation, organs morphogenesis, cellular fusion and longevity. Moreover, Gal-3 is involved in the chemoattraction of monocytes/macrophages, its expression is increased when monocytes differentiate into macrophages, and it is crucial for the monocyte-monocyte interaction. These characteristics are considered important for chronic inflammatory processes, as well as for the mechanisms of cardiac fibrosis driving adverse cardiac remodeling [46,47,48]. Of note, Gal-3 has been associated with other cardiovascular diseases, but also with many other inflammatory/chronic conditions, such as infections, gastritis, liver fibrosis and cancers [49]. Although several factors such as comorbidities, chronic inflammation and metabolic abnormalities have been identified as potential mechanisms linked to frailty, the pathophysiology of frailty in HF patients is still not clear. Senescent cells and their bioactive compounds, which include inflammatory cytokines, growth factors and proteases, have an important impact on the chronic inflammatory environment and acceleration of aging [14,50]. Different studies performed in vitro or experimental models report that cellular senescence is dependent on Gal-3 signaling pathways [51,52]. Therefore, we suggest that the up-regulation of Gal-3 associated with HF development might interfere with mechanisms of premature senescence, accelerated aging and frailty development. A recent study, performed in a large population of elderly subjects, revealed that Gal-3 is associated with an impaired myocardial function in physiological ageing, describing a potential role in an early (preclinical) phase of myocardial ageing [20]. The role of Gal-3 as an independent predictor of outcomes in HF has not been confirmed; however, adding Gal-3 to the well-established NT-proBNP ameliorated the prediction of adverse outcomes [53], probably because the population of patients with higher Gal-3 were also the most-frail. Furthermore, Testa and colleagues [54] reported that a Gal-3 value >17.6 ng/mL added to the plasma BNP level > 500 pg/mL significantly improved the predictive mortality in elderly with chronic HF, and additionally the disability level was higher. 

In our study, we described a significant and independent association of Gal-3 with frailty in HF patients. Although we applied a multivariate approach, adjusting our results for several HF characteristics/parameters, including the NYHA functional class, LVEF and NT-proBNP levels, the association between Gal-3 and frailty was not attenuated and came out as significant and independent. Finally, the application of a decision curve analysis model revealed that the inclusion of Gal-3 in the clinical decision-making chain would improve patient management. 

Study limitations: This is a single center experience with a relatively small group of patients, and, therefore, it deserves further confirmation in a future larger HF population, which would also allow an external validation of the present results. Moreover, we only performed a single point determination of Gal-3 serum levels, whereas several Gal-3 determinations would probably be of interest in providing additional information with regard to its association with frailty. 87.5% of the population that was included were males, and we could not exclude a gender bias in our study. Finally, an assessment of frailty with other more comprehensive scales will be necessary in order to better define this association.

## 5. Conclusions

This study identified Gal-3 serum levels as a circulating biomarker independently associated with frailty, in a population of elderly systolic HF patients. Our findings may open the way for future investigations to better define the role of Gal-3 in the pathophysiology of frailty in HF and its utility as a biomarker in the prognosis of frail populations.

## Figures and Tables

**Figure 1 jcm-09-02229-f001:**
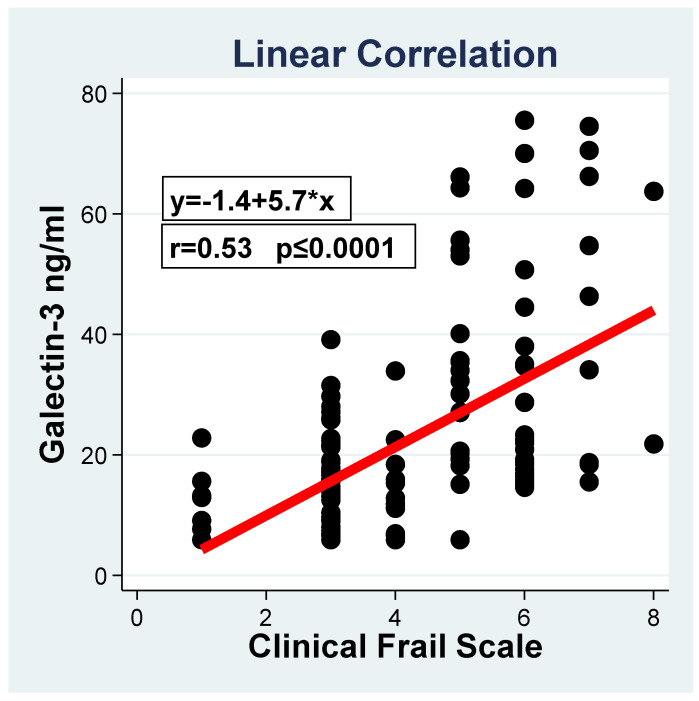
Linear correlation between the Galectin-3 and Clinical Frail values.

**Figure 2 jcm-09-02229-f002:**
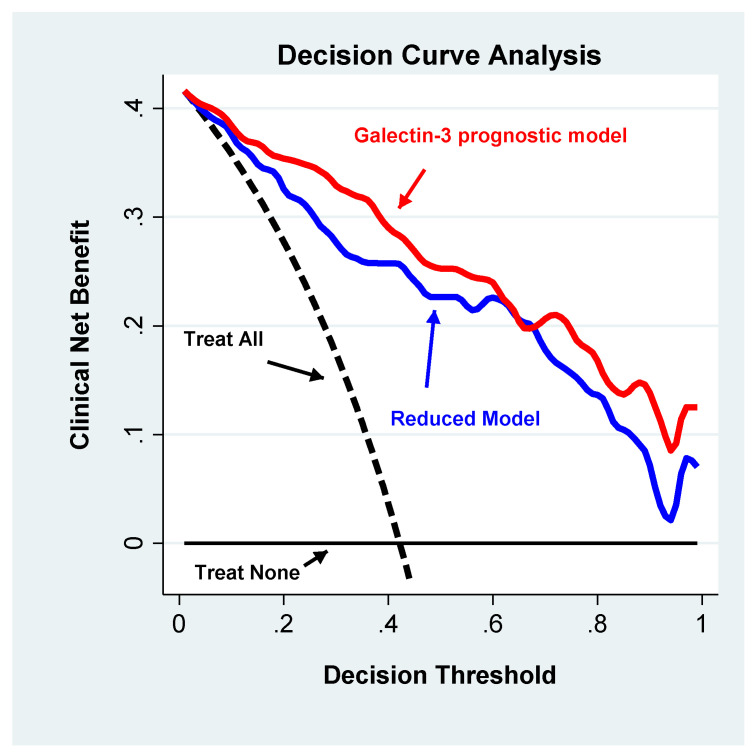
Decision Curve Analysis. The “treat none” (black continuous line) and “treat all’ (black continuous line) curves are compared with the net benefit curves of the two multivariate models: Galectin-3 prognostic model (red continuous line) and partial model without Galectin-3 (blue continuous line). The full model profile is higher than the partial model profile across the critical range of the decision threshold probabilities (20–60%). Both models show curves above those of the “treat none” and “treat all” strategies.

**Table 1 jcm-09-02229-t001:** Characteristics of patients in the overall study population, stratified as frail and non-frail.

Characteristics	All Population (N = 128)	Frail (N = 54)	Non- Frail (N = 74)	*p*-Value
Age, years	69.2 ± 4.8	70.5 ± 5.4	68.2 ± 4.2	0.008
Gender M, N (%)	112 (87.5)	45/54 (83.4)	66/73 (90.4)	0.28
BMI, kg/m^2^	25.4 ± 4.3	24.5 ± 4.6	26.0 ± 4.2	0.07
WBC/µL,	7833 ± 2623.5	8104.7 ± 4145.9	7430.6 ± 2099.6	0.78
Hbg g/dL	12.8 ± 1.5	12.2 ± 1.6	13.6 ± 1.7	0.001
Fibrinogen	400.2 ± 65.8	405.2 ± 112.4	394.8 ± 108.3	0.75
Na mmol/L	138.4 ± 3.2	138.3 ± 4.5	138.7 ± 3.8	0.68
GFR mL/kg	65.2 ± 18.4	59.1 ± 20.4	70.2 ± 14.8	0.001
CRP mg/L	7.9 ± 10.7	13.4 ± 13.8	3.7 ± 4.2	<0.0001
LVEF %	28.7 ± 8.5	26.7 ± 6.1	30.2 ± 10.2	0.02
NT proBNP pg/mL	5922.4 ± 15,099.9	11,427.9 ± 21,803.4	1856.4 ± 3570.1	0.002
Galectin-3 ng/mL	22.8 ± 16.9	34.4 ± 19.3	14.3 ± 7.6	<0.0001
Hypertension, N (%)	80 (62.5)	37/54 (68.5)	43/74 (58.1)	0.23
Dyslipidemia, N (%)	108 (84.3)	43/54 (79.6)	65/74 (87.8)	0.21
NYHA class III,IV N(%)	58 (54.7)	35/54 (64.8)	23/74 (31.1)	<0.0001
CIRS-CI	3.73 ± 2.2	4.6 ± 2.2	3.1 ± 1.9	<0.0001
CFS	4.27 ±1.7	5.9 ± 0.8	3.1 ± 0.8	<0.0001
ACEInhib/ARBs	104 (81.2)	41/54 (75.9)	63/74 (85.1)	0.19
Beta-blockers	85 (66.4)	40/54 (74.1)	45/74 (60.8)	0.12
Diuretics *	112 (87.5)	48/54 (88.9)	64/74 (86.4)	1.0
Drugs Number	4.84 ± 1.4	4.9 ± 1.2	4.8 ± 1.4	0.34

BMI: Body Mass Index; WBC: White Blood Cells; Hbg: Haemoglobin; GFR: Glomerular Filtration Rate; CRP: C Reactive Phase Protein; LVEF: Left Ventricular Ejection Fraction; NT proBNP: N-terminal -pro-Brain Natriuretic Peptide NYHA: New York Heart Association; CIRS-CI: Cumulative Illness Rating Scale Comorbidity Index; CFS: Clinical Frailty Scale; ACEinhb.: ACE inhibitors; ARBs: Angiotensin Receptor Blockers. * Diuretics and or Aldosterone Antagonists.

**Table 2 jcm-09-02229-t002:** Multivariable logistic regression models.

	Global Pseudo R^2^ * = 0.5			
Variables	Odd’s Ratio	95% CI	*p*-Value	Partial Contribution to Global R^2^
Age (decades)	3.29	1.03–10.55	0.045	6.3%
Gender	0.86	0.19–4.03	0.854	NA
BMI	0.95	0.83–1.10	0.415	NA
CKD	1.47	0.34–6.36	0.605	NA
CIRS-CI (SD units)	1.85	1.03–3.32	0.039	10.3%
NYHA Class III, IV	1.53	0.46–5.13	0.456	NA
LVEF	0.93	0.84–1-03	0.172	NA
NT-proBNP (SD units)	2.39	1.22–4.73	0.012	24.5%
Hgb	0.95	0.64–1.40	0.828	NA
CRP (SD units)	3.73	1.24–11.22	0.019	19.5%
Gal-3 (SD units)	5.65	1.97–16.22	0.019	39.4%

BMI: Body Mass Index; CKD: Chronic Kidney Disease defined as GFR ≤ 50 mL/kg/m^2^; CIRS: Standardized Cumulative Illness Rating Scale-Comorbidity Index; NYHA: New York Heart Association Class; LVEF: Left Ventricular Ejection Fraction; NT-proBNP: N-terminal -pro-Brain Natriuretic Peptide lnNT-proBNP: Standardized logarithmic transformation of NT-proBNP; Hgb: Haemoglobin; CRP: Standardized C-Reactive Phase Protein; Gal-3: Standardized Galectin three, NA: Not Applicable. * Global pseudo R2 model including: age, CIRS, lnNT-proBNP, CRP and Gal-3.

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
