# Peer review of "Impact of Galectin-3 Circulating Levels on Frailty in Elderly Patients with Systolic Heart Failure"

_jcm, 2020, doi:10.3390/jcm9072229_

Round 1

Reviewer 1 Report

This study reported the association of Gal-3 and frailty in elderly patients with HF. However, there is a lack of logical explanation for the need for research. The conclusion was overstated. The authors need to revise and clarify below points:

1) In the conclusion of abstract, the authors need to focus on Gal-3 like title.

2) In Introduction, there is a lack of logical explanation for the need for research. Previous studies have already shown that Gal-3 is related to inflammatory condition and the level of Gal-3 increased in patients with HF. And there are already other biomarkers (CRP, NT-proBNP) in patients with HF and frailty. But the study purpose is written without explaining why to check Gal-3. Please clarify this in introduction.

3) In method, please explain why you focus on patients with HF with systolic dysfunction.

4) In method, please explain why you assess the other biomarkers (CRP, NT-proBNP) with Gal-3. I think there is a reason to confirm these biomarkers together, so it is necessary to specify the reason in this manuscript. And please explain how you measured these biomarkers in 2.3. section.

5) Please add the approval number of institutional review board.

6) In results, please present only the important results with numerical statistics and do not repeat the numbers in the table.

7) The main finding is that Gal-3 is significantly increased in frail patients with HF. This is important and meaningful because this can explain the pathophysiological link between frailty and patients with HF. But the conclusion of this study focused on the use of Gal-3 to identify frailty in patients with HF. Many valid tools have already been reported to quickly identify frailty in the clinical situations. Therefore, this conclusion is not appropriate because it means that physician have to wait for Gal-3 level in blood to confirm frailty in a busy clinical environment.

8) According to the title and study purpose, the discussion of the relationship between Gal-3 and frailty should be led. Explaining the relationship between NT-proBNP and frailty in detail in discussion is outside the purpose of study.

9) It is necessary to explain in detail the clinical application of the DCA model, which is the main result. In discussion, only “Finally, the application of decision curve analysis model revealed that inclusion of Cal-3 in the clinical decision-making chain would improve patient management.” is written. However, it is to properly discuss the results of this study by fully explaining the meaning of this sentence and describing examples of how to apply it specifically in the clinical setting.

10) Please check out the typos and missing words (ex. LVEF in 2 page).

Reviewer 2 Report

In the submitted manuscript, Komici et al. discussed that Galectin-3 (Gal-3) is possible biomarker for frailty in old patients with systolic heart failure (HF). All the parts of the manuscript have been explained very carefully and data were discussed effectively. The manuscript did find, for the first time, an independent correlation of frailty and Gal-3 by excluding patients with chronic inflammatory disorders. Clinical Frailty Scale (CFS) score was also appropriate and in-line with previously published report. Although manuscript has some promise, the main idea that higher Gal-3 levels correspond to higher level of frailty, that is, functional impairment has been discovered and reported by Testa et al. (https://pubmed.ncbi.nlm.nih.gov/30262779/) In this published report, authors did not measure frailty via CFS like Komici and colleagues, however, they did measure Barthel Index which I found had been used in few publications to measure frailty.

In summary, the independent relation of Gal-3 and frailty is the main novelty of the submitted manuscript but higher Gal-3 higher frailty relationship (may be dependent) is already published. 

Reviewer 3 Report

In this manuscript, Komici et al., aimed to investigate the potential of Galectin-3 to serve as a biomarker of frailty in heart failure patients. By using multivariable regression analysis and decision curve analysis, the authors identified that Gal-3 was among the significant predictors of frailty along with CRP, NT-proBNP.

In general, this study was well designed, and the manuscript was well written. The results shown significance for the identifying novel biomarker to evaluation of frailty in HF patients. However, there are still some points need to be revised:

  1. I suggest the authors to add more info in introduction/discussion part on the weakness of current tools in evaluating frailty, e.g. what’s the false negative rate of the current methods/biomarkers? How would this weakness affect the outcome of HF patient care and prognosis?
  2. Have frailty been defined in a progressive way, e.g. early stage, medium stage, late stage? If so, what do the patients in this cohort contribute in each stage? Which stage does Gal-3 correlate to in this case?
  3. As the majority of the cohort is male (87.5%), I suggest that the authors add this in the limitation part for the possible gender bias of the results.

Round 2

Reviewer 1 Report

The authors answered all of my questions well.

Thank you for your response.

Reviewer 2 Report

Authors did answer the reason for rejection very clearly and precisely. I kindly request the authors to cite the work by Testa et al. (https://pubmed.ncbi.nlm.nih.gov/30262779/), if appropriate.

Having said that, I am really happy to see the overall improved quality of the paper.